# COMPOSITIONAL VIDEO SYNTHESIS WITH ACTION GRAPHS

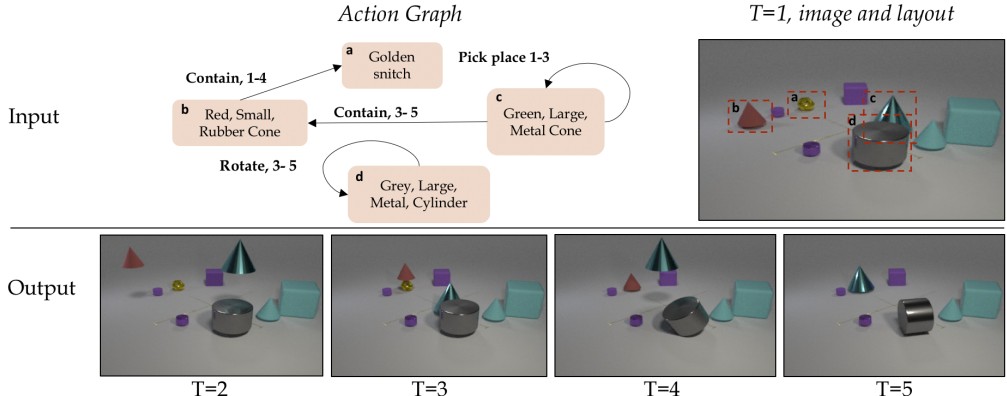

Figure 1: We focus on video synthesis from actions and propose a new task called *Action Graph to Video*. To represent input actions, we use a graph structure called *Action Graph*, and together with the first frame and first scene layout, our goal is to synthesize a video that matches the input actions. For illustration, we include above a (partial) example. Our model outperforms various baselines and can generalize to previously unseen compositions of actions.

## ABSTRACT

Videos of actions are complex signals, containing rich compositional structure. Current video generation models are limited in their ability to generate such videos. To address this challenge, we introduce a generative model (AG2Vid) that can be conditioned on an Action Graph, a structure that naturally represents the dynamics of actions and interactions between objects. Our AG2Vid model disentangles appearance and position features, allowing for more accurate generation. AG2Vid is evaluated on the CATER and Something-Something datasets and outperforms other baselines. Finally, we show how Action Graphs can be used for generating novel compositions of actions.

## 1 INTRODUCTION

Learning to generate visual content is a fundamental task in computer vision, with numerous applications from sim-to-real training of autonomous agents, to creating visuals for games and movies. While the quality of generating still images has leaped forward recently (Karras et al., 2020; Brock et al., 2019), generating videos is much harder. Generating actions and interactions is perhaps the most challenging aspect of conditional video generation. Actions create long-range spatio-temporal dependencies between people and the objects they interact with. For example, when a player passes a ball, the entire movement sequence of all entities (thrower, ball, receiver) must be coordinated and carefully timed. The current paper focuses on this difficult obstacle, the task of generating coordinated and timed actions, as an important step towards generating videos of complex scenes.

Current approaches for conditional video generation are not well suited to condition the generation on actions. First, *future video prediction* (Ye et al., 2019; Watters et al., 2017), generates future frames based on an initial input frame, but a first frame cannot be used to predict coordinated actions. Second, in video-to-video , the goal is to translate a sequence of semantic masks into an output video.

However, segmentation maps contain only class information, and thus do not explicitly capture the action information. As Wang et al. (2018a) notes, this is a limitation that leads to systematic mistakes, such as in the case of car turns. Finally, text-to-video (Li et al., 2018; Gupta et al., 2018) is potentially useful for generating videos of actions because language can describe complex actions. However, in applications that require a precise description of the scene, language is not ideal due to ambiguities (MacDonald et al., 1994) or subjectivity of the user (Wiebe et al., 2004). Hence, we address this problem with a more structural approach.

To provide a better way to condition on actions, we introduce a formalism we call an "Action Graph" (AG), propose a new task of "Action Graph to Video" (AG2Vid), and present a model for this task. An AG is a graph structure aimed at representing coordinated and timed actions. Its nodes represent objects, and edges represent actions annotated with their start and end time (Fig. 1). We argue that AGs are an intuitive representation for describing timed actions and would be a natural way to provide precise inputs to generative models. A key advantage of AGs is their ability to describe the dynamics of object actions precisely in a scene.

In our AG2Vid task, the input is the initial frame of the video and an AG. Instead of generating the pixels directly, our AG2Vid model uses three levels of abstraction. First, we propose an action scheduling mechanism we call "Clocked edges" that tracks the progress of actions in different timesteps. Second, based on this, a graph neural network (Kipf & Welling, 2016) operates on the AGs and predicts a sequence of scene layouts, and finally, pixels are generated conditioned on the predicted layouts. We apply this AG2Vid model to the CATER (Girdhar & Ramanan, 2020) and Something-Something (Goyal et al., 2017) datasets and show that this approach results in realistic videos that are semantically compliant with the input AG. To further demonstrate the expressiveness of AG representation and the effectiveness of the AG2Vid model, we test how it generalizes to previously unseen compositions of the learned actions. Human raters then confirm the correctness of the generated actions.[1]

Our contributions are as follows: 1) Introducing the formalism of *Action Graphs* (AG) and proposing a new video synthesis task. 2) Presenting a novel action-graph-to-video (AG2Vid) model for this task. 3) Using the AG and AG2Vid model, we show this approach generalizes to the generation of novel compositions of the learned actions.

## 2 RELATED WORK

Video generation is challenging because videos contain long range dependencies. Recent generation approaches (Vondrick et al., 2016; Kumar et al., 2020; Denton & Fergus, 2018; Lee et al., 2018; Babaeizadeh et al., 2018; Villegas et al., 2019) extended the framework of unconditional image generation to video, based on a latent representation. For example, MoCoGAN (Tulyakov et al., 2018) disentangles the latent space representations of motion and content to generate a sequence of frames using RNNs; TGAN (Saito et al., 2017) generates each frame in a video separately while also having a temporal generator to model dynamics across the frames. Here, we tackle a different problem by aiming to generate videos that comply with AGs.

Conditional video generation has attracted considerable interest recently, with focus on two main tasks: video prediction (Mathieu et al., 2015; Battaglia et al., 2016; Walker et al., 2016; Watters et al., 2017; Kipf et al., 2018; Ye et al., 2019) and video-to-video translation (Wang et al., 2019; Chan et al., 2019; Siarohin et al., 2019; Kim et al., 2019; Mallya et al., 2020). In prediction, the goal is to generate future video frames conditioned on few initial frames. For example, it was proposed to train predictors with GANs (Goodfellow et al., 2014) to predict future pixels (Mathieu et al., 2015). However, directly predicting pixels is challenging (Walker et al., 2016). Instead of pixels, researchers explored object-centric graphs and perform prediction on these (Battaglia et al., 2016; Luc et al., 2018; Ye et al., 2019). While inspired by object-centric representations, our method is different from these works as our generation is goal-oriented, guided by an AG. The video-to-video translation task was proposed by Wang et al. (2018a), where a natural video was generated from frame-wise semantic segmentation annotations. However, densely labeling pixels for each frame is expensive, and might not even be necessary. Motivated by this, researchers have sought to perform generation conditioned on more accessible signals including audio or text (Song et al., 2018; Fried

---

[1]Our code and models will be released upon acceptance.

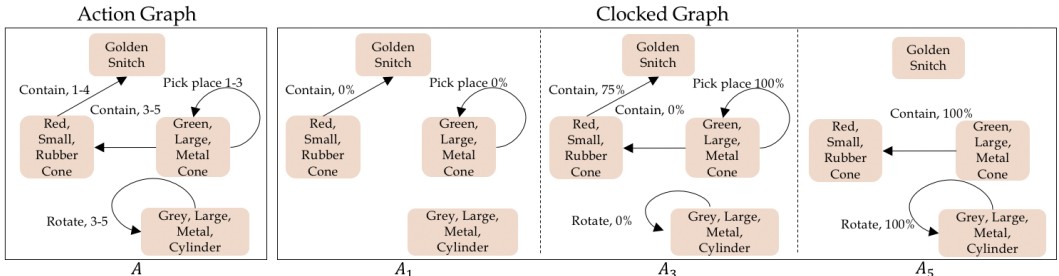

Figure 2: Example of a partial Action Graph execution schedule in different time-steps.

et al., 2019; Ginosar et al., 2019). Here, we propose to synthesize videos conditioned on a novel AG, which is easy to obtain compared to semantic segmentation and is a more structured representation compared to natural audio and text.

A recent method, HOI-GAN (HG) (Nawhal et al., 2020b), was proposed for the generation task of a single action and object. Specifically, this work addresses the zero-shot setting, and the model is tested on action and object compositions which are first presented at test time. Our focus is on generation of multiple simultaneous actions over time, performed by multiple objects. Our approach directly addresses this challenge via the AG representation and the notion of clocked edges.

Various methods have been proposed to generate videos based on input text (Marwah et al., 2017; Pan et al., 2017; Li et al., 2018). Most recent methods typically used very short captions which do not contain complex descriptions of actions. For example, (Li et al., 2018) used video-caption pairs from YouTube, where typical captions are "playing hockey" or "flying a kite". Gupta et al. (2018) proposed the Flinstones animated dataset and introduced the CRAFT model for text-to-video generation. While the CRAFT model relies on text-to-video retrieval, our approach works in an end-to-end manner and aims to accurately synthesize the given input actions.

Scene Graphs (SG) (Johnson et al., 2015; 2018) are a structured representation that models scenes, where objects are nodes and relations are edges. SGs have been widely used in various tasks including image retrieval (Johnson et al., 2015; Schuster et al., 2015), relationship modeling (Krishna et al., 2018; Schroeder et al., 2019; Raboh et al., 2020), SG prediction (Xu et al., 2017; Newell & Deng, 2017; Zellers et al., 2018; Herzig et al., 2018), and image captioning (Xu et al., 2019). Recently, SGs have been applied to image generation (Johnson et al., 2018; Deng et al., 2018; Herzig et al., 2020), where the goal is to generate a natural image corresponding to the input SG. More generally, spatio-temporal graphs have been explored in the field of action recognition (Jain et al., 2016; Sun et al., 2018; Wang & Gupta, 2018; Yan et al., 2018; Girdhar et al., 2019; Herzig et al., 2019; Materzynska et al., 2020). For example, a space-time region graph is proposed by (Wang & Gupta, 2018) where object regions are taken as nodes and a GCN (Kipf & Welling, 2016) is applied to perform reasoning across objects for classifying actions. Recently, it was also shown by (Ji et al., 2019; Yi et al., 2019; Girdhar & Ramanan, 2020) that a key obstacle in action recognition is the ability to capture the long-range dependencies and compositionality of actions. While inspired by these approaches, we focus on generating videos which is a very different challenge.

Recently, Ji et al. (2019) presented Action Genome, a new video dataset annotated by SGs. This dataset includes spatio-temporal SG annotations, where for each video, few individual frames were chosen and spatially annotated by SGs. Here, we use the Something-Something V2 (Goyal et al., 2017) dataset that is larger (200K vs. 10K videos) and more diverse since it includes basic human activities created by a large number of crowd workers. Finally, we propose the Action Graph representation, which we view as a temporal extension of SGs.

## 3 ACTION GRAPHS

Our goal in this work is to build a model for synthesizing videos that contain a specified set of actions. A key component in this effort is developing a semantic representation to describe the actions performed by different objects in the scene. Towards this end, we introduce a formalism we call *Action Graph* (AG). In an AG, nodes correspond to objects, and edges correspond to actions that

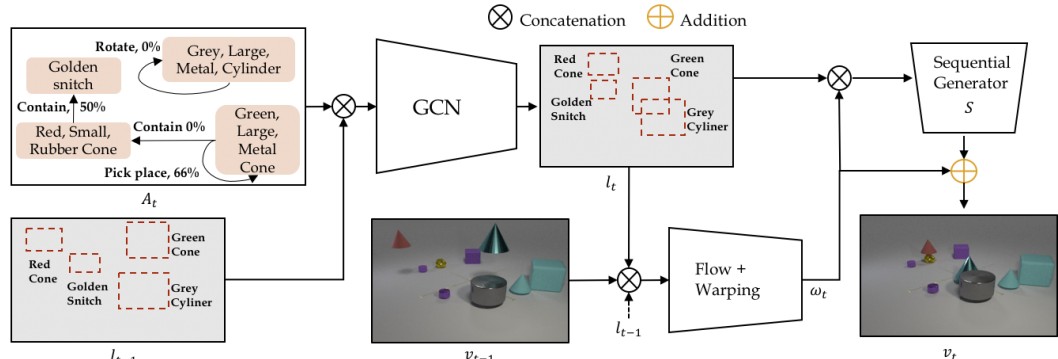

Figure 3: Our AG2Vid Model. The AG $A_t$ describes the execution stage of each action at time $t$. Together with the previous layout $\ell_{t-1}$, it is used to generate the next layout $\ell_t$ which has object representations that are enriched with $A_t$ actions information. Then, $\ell_t, \ell_{t-1}, v_{t-1}$ are used to generate the next frame.

these objects participate in. Objects and actions are annotated by semantic categories, and actions are also annotated by their start and end times.

More formally, an AG is a tuple $(\mathcal{C}, \mathcal{A}, O, E)$ described as follows:

- **An alphabet of object categories $\mathcal{C}$.** Categories can be compounded and include attributes. For example "Blue Cylinder" or "Large Box".

- **An alphabet of action categories $\mathcal{A}$.** For Example "Slide" and "Rotate". Similarly, actions can contain attributes (e.g., rotation speed).

- **Object nodes O:** A set $O \in \mathcal{C}^n$ of $n$ objects.

- **Action edges E:** Actions are represented as labeled directed edges between object nodes. Each edge is annotated with an action category and with the time period during which the action is performed. Formally, each edge is of the form $(i, a, j, t_s, t_e)$ where $i, j \in \{1, ..., n\}$ are object instances, $a \in \mathcal{A}$ is an action and $t_s, t_e \in \mathbb{N}$ are action start and end time. Thus, this edge implies that object $i$ (which has category $o_i$) performs an action $a$ over object $j$, and that this action takes place between times $t_s$ and $t_e$. We note that an AG edge can directly model actions over a single object and a pair of objects. For example, "Swap the positions of objects $i$ and $j$ between time 0 and 9" is an action over two objects corresponding to edge $(i, swap, j, 0, 9)$. Some actions, such as "Rotate", involve only one object and will therefore be specified as self-loops.

## 4 ACTION GRAPH TO VIDEO VIA CLOCKED EDGES

We now turn to the key challenge of this paper: transforming an AG into a video. Naturally, this transformation will be learned from data. The generation problem is defined as follows: we wish to build a generator $G$ that takes as input an AG and outputs a video. We will also allow conditioning on the first video frame and layout, so we can preserve the visual attributes of the given objects.[2]

There are multiple unique challenges in generating a video from an AG that cannot be addressed using current generation methods. First, each action in the graph unfolds over time, so the model needs to "keep track" of the progress of actions rather than just condition on previous frames as commonly done. Second, AGs may contain multiple concurrent actions and the generation process needs to combine them in a realistic way. Third, one has to design a training loss that captures the spatio-temporal video structure to ensure that the semantics of the AG is accurately captured.

**Clocked Edges.** As discussed above, we need a mechanism for monitoring the progress of action execution during the video. A natural approach is to keep a "clock" for each action, for keeping track of action progress as the video progresses. See Fig. 2 for an illustration. Formally, we keep a clocked version of the graph $A$ where each edge is augmented with a temporal state. Let $e =$

---

[2]Using the first frame and layout can be avoided by using a SG2Image model (Johnson et al., 2018; Ashual & Wolf, 2019; Herzig et al., 2020) for generating the first frame.

$(i, a, j, t_s, t_e) \in E$ be an edge in the AG $A$. We define the progress of $e$ at time $t$ to be $r = \frac{t-t_s}{t_e-t_s}$, and clip $r$ to $[0, 1]$. Thus, if $r = 0$ the action has not started yet, if $r \in (0, 1)$ it is currently being executed, and if $r = 1$ it has completed. We then create an augmented version of the edge $e$ at time $t$ given by $e_t = (i, a, j, t_s, t_e, r)$. We define $A_t = \{e_t | e \in A\}$ to be the AG at time $t$. To summarize the above, we take the original graph $A$ and turn it into a sequence of AGs $A_0, \ldots, A_T$, where $T$ is the last time-step. Each action edge in the graph now has its unique clock for its execution. This facilitates both a timely execution of actions and coordination between actions.

## 4.1 THE AG2VID MODEL

Next, we describe our proposed AG-to-video model (AG2Vid). Fig. 3 provides a high-level illustration of our model architecture. The rationale of our generation process is that first the AG is used to produce intermediate layouts, and then these layouts are used to produce frame pixels We let $\ell_t = (x_t, y_t, w_t, h_t, z_t)$ denote the set of predicted layouts for all the $n$ objects in the video at time $t$. The values $x_t, y_t, w_t, h_t \in [0, 1]^n$ are the bounding box coordinates for all objects, and $z_t$ is a descriptor vector for the object (later used for frame generation). Let $v_t$ denote the generated frame at time $t$, and $p(v_2, \ldots, v_T, \ell_2, \ldots \ell_T | A, v_1, \ell_1)$ denote the generating distribution of the frames and layouts given the AG and the first frame $v_1$ and scene layout $l_1$.

We assume that the generation of the frame and layout directly depends only on recent generated frames and layouts.[3] Specifically, this corresponds to the following form for $p$: $p(v_2, ..., v_T, \ell_2, ..., \ell_T | A, v_1, l_1) = \prod_{t=2}^{T} p(\ell_t | A_t, \ell_{t-1}) p(v_t | v_{t-1}, \ell_t, \ell_{t-1})$. We refer to the distribution $p(l_t|\cdot)$ as *The Layout Generating Function (LGF)* and to $p(v_t|\cdot)$ as *The Frame Generating Function (FGF)*. Next, we describe how we model these distributions as functions.

**The Layout Generating Function (LGF).** At time $t$ we want to use the previous layout $\ell_{t-1}$ and current AG $A_t$ to predict the current layout $\ell_t$. The rationale is that $A_t$ captures the current state of the actions and can thus "propagate" $\ell_{t-1}$ to the next layout. This prediction requires integrating information from different objects as well as the progress of the actions given by the edges of $A_t$. Thus, a natural architecture for this task is a *Graph Convolutional Network* (GCN) that operates on the graph $A_t$ whose nodes are "enriched" with the layouts $\ell_t$. Formally, we construct a new graph of the same structure as $A_t$, with new features on nodes and edges. At the graph node corresponding to object $i$ the features are comprised of the previous object location defined in $\ell_{t-1}^i$ and object class embedding. The features on the edges are the embedding of action $a$ and the progress of the action $r$, taken from $(i, a, j, r)$ from $A_t$. Then, node and edge features are repeatedly re-estimated for $K$ steps using a GCN. The resulting activations of the $i$th object at timestep $t$ are $z_t^i \in \mathbb{R}^D$ which we use as the new object descriptor. An MLP is then applied to it to produce the new box coordinates, which together form the predicted layout $\ell_t$. For more details refer to Sec.1 in the Suppl.

**The Frame Generating Function (FGF).** After obtaining the layout $\ell_t$ which contains updated objects representations $z_t$, we wish to use it along with $v_{t-1}$ and $\ell_{t-1}$ to predict the next frame. The idea is that $\ell_t, \ell_{t-1}$ characterize how objects should move, $z_t, z_{t-1}$ should capture the object-actions dynamics, and $v_{t-1}$ shows their last physical appearance. Combining these information sources we should be able to generate the next frame accurately. As a first step, we construct a mask $m_{t-1}, m_t \in \mathbb{R}^{H \times W \times D}$ using the embedding and layout pairs. Then, we estimate the optical flow at time $t$, denoted by $f_t$. We let $f_t = F(v_{t-1}, m_{t-1}, m_t)$. The idea is that given the previous frame and two consecutive layouts, we should be able to predict in which direction pixels in the image will move, namely predict the flow. The flow prediction network $F$ is similar to (Ilg et al., 2017), and it is trained using an auxiliary loss and does not require additional supervision (see section 4.2). Given the flow $f_t$ and previous frame $v_{t-1}$ a natural estimate of the next frame is to use a warping function (Zhou et al., 2016) $w_t = W(f_t, v_{t-1})$. Finally we fine-tune $w_t$ via a network $S(m_t, w_t)$ that provides an additive correction resulting in the final frame prediction: $v_t = w_t + S(m_t, w_t)$, where the $S$ network a SPADE generator (Park et al., 2019).

---

[3]Note that layout contain descriptors that can encode all generation history.

Figure 4: Qualitative examples of generation on CATER and Something Something. AG2Vid generated videos of four and eight standard actions on CATER and Something Something, respectively. For CATER we also used AGs with multiple simultaneous actions, and the generated actions indeed correspond to those (verified manually). For more examples please refer to Figure 1 and 2 in the Supp. *Click the image to play the video clip in a browser*.

## 4.2 LOSSES AND TRAINING

We use ground truth frames $v_t^{GT}$ and layouts $\ell_t^{GT}$ for training,[4] and use the following losses:

**Layout Prediction Loss $\mathcal{L}_\ell$.** Defined as $\mathcal{L}_\ell = \|\ell_t - \ell_t^{GT}\|_1$, the $L_1$ loss between ground-truth bounding boxes $\ell_t^{GT}$ and predicted boxes $\ell_t$. Here we ignore the object descriptor part of $\ell_t$.

**Pixel Action Discriminator Loss $\mathcal{L}_A$.** For the generated pixels $v_t$ we employ a GAN loss that uses a discriminator between generated frames $v_t$ and GT frames $v_t^{GT}$ conditioned on $A_t$ and $l_t$. Formally, let $D_A$ be a discriminator with output in $(0, 1)$. First, a GCN is applied over $A_t$ to obtain objects representations, which are then using together with the GT layout boxes to construct a scene layout. The layout and frames are then concatenated and fed into a multi-scale PatchGAN discriminator (Wang et al., 2018b). The loss is then the GAN loss (e.g., see Isola et al. (2017)):

$$\mathcal{L}_A = \max_{D_A} \mathbb{E}_{GT} \left[ \log D_A(A_t, v_t^{GT}, \ell_t^{GT}) \right] + \mathbb{E}_p \left[ \log(1 - D_A(A_t, v_t, \ell_t^{GT})) \right] \tag{1}$$

where $GT$ corresponds to sampling frames from the ground truth videos, and $p$ corresponds to sampling from the generated videos. Optimization of this loss is done in the standard way of alternating gradient ascent on $D_A$ parameters and descent on generator parameters.

**Flow Loss $\mathcal{L}_f$.** The flow loss measures the error between the warps of the previous frame and the ground truth of the next frame $v_t^{GT}$: $\mathcal{L}_f = \frac{1}{T-1} \sum_{t=1}^{T-1} \|w_t - v_t\|_1$, where $w_t = W(f_t, v_{t-1})$ as defined in Section 4.1. This loss was proposed previously by (Zhou et al., 2016; Wang et al., 2018a).

**Perceptual and Feature Matching Loss $\mathcal{L}_P$.** We use these losses as proposed in pix2pixHD (Wang et al., 2018b; Larsen et al., 2016) and other previous works.

The overall optimization problem is to minimize the weighted sum of the losses.

## 5 EXPERIMENTS AND RESULTS

We evaluate our AG2Vid model on two datasets: *CATER* and *Something Something V2* (Smth). For each dataset, we learn an AG2Vid model with a given set of actions. We then evaluate the visual quality of the generated videos and measure how they semantically comply with the input actions. Last, we estimate the generalization of the AG2Vid model to novel composed actions.

**Datasets.** We use two datasets: **(1) CATER** (Girdhar & Ramanan, 2020) is a synthetic video dataset originally created for action recognition and reasoning. Each video contains multiple objects performing actions. The dataset contains bounding-box annotations for all objects, as well as labels of the actions. These include: "Rotate", "Cover", "Pick Place" and "Slide". See Figure 7 for

---

[4]GT layouts can be obtained automatically from videos using object tracking as a pre-processing step.

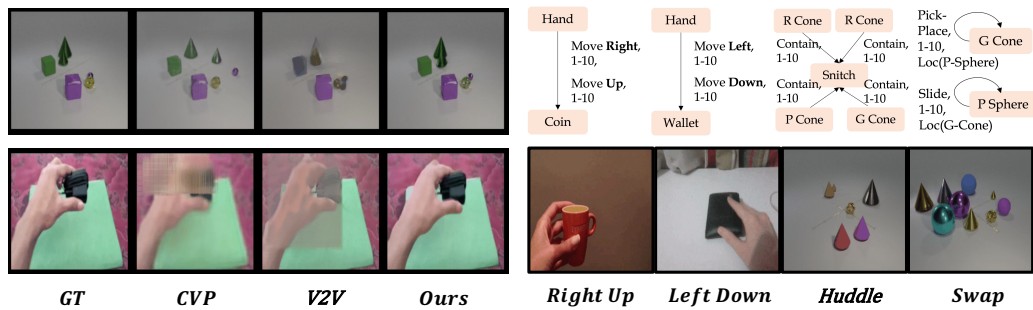

| GT | CVP | V2V | Ours | | Right Up | Left Down | Huddle | Swap |

Figure 5: Comparison of baselines methods. The top row are based on *CATER* videos, while the bottom row are based on *Something Something*. *Click the images to play the video clips in a browser.*

Figure 6: Composing unseen actions in *Something-Something* and *CATER*. For example, the "swap" action is composed by combining the "Pick-Place" and "Slide" actions on frames $1 - 10$ and their locations.

Table 1: **Human evaluation** of action generation with respect to Semantic Accuracy and Visual Quality. For each metric, raters selected the better of two generation methods. In the results $XX/YY$ means that AG2Vid was selected as better for $X\%$ of the presented pairs. Image resolution is $256 \times 256$.

| Methods | Semantic Accuracy | | Visual Quality | |
| AG2Vid / Baseline | CATER | Smth | CATER | Smth |
| --- | --- | --- | --- | --- |
| AG2Vid / CVP (Ye et al., 2019) | **85.7**/14.3 | **90.6**/9.4 | **76.2**/23.8 | **93.8**/6.2 |
| AG2Vid / HG (Nawhal et al., 2020a) | -/− | **84.6**/15.4 | -/− | **88.5**/11.5 |
| AG2Vid / V2V (Wang et al., 2018a) | **68.8**/31.2 | **84.4**/15.6 | **68.8**/31.2 | **96.9**/3.1 |
| AG2Vid / RNN | **56.0**/44.0 | **80.6**/19.4 | **52.0**/48.0 | **77.8**/22.2 |
| AG2Vid / AG2Vid, GTL | 48.6/**51.4** | 46.2/**53.8** | 42.9/**57.1** | 50.0/50.0 |

qualitative examples. For "Pick Place" and "Slide" we include the action destination coordinates. We use these actions to create action graphs for training and evaluation.

We employ the standard CATER training partition (3849 videos) and split the validation into 30% val (495 videos) and use the rest for testing (1156 videos). **(2) Something Something V2 (Goyal et al., 2017)** contains real world videos of humans performing basic actions. Here we use the eight most frequent actions (e.g., "Putting [something] on a surface" and "Covering [something] with [something]"). All videos contain up to three different objects, including the hand which is performing the action. We use the box annotations of the objects from Materzynska et al. (2020). See Sec. 3 in Suppl for the full list of actions.

**Implementation details.** The GCN model uses $K = 3$ hidden layers and an embedding layer of 128 units for each object and action. For optimization we use ADAM Kingma & Ba (2014) with $lr = 1e - 4$ and $(\beta_1, \beta_2) = (0.5, 0.99)$. Models were trained on an NVIDIA V100 GPU. For loss weights (see section 4.2) we use $\lambda_B = \lambda_F = \lambda_P = 10$ and $\lambda_A = 1$. For training we use a batch size of 2. We use videos of 8 FPS and 6 FPS for CATER and Smth and evaluate on videos consisting of 16 frames which correspond to spans of 2.7 and 2 seconds accordingly.

**Performance metrics.** The AG2Vid outputs can be quantitatively evaluated as follows. **a) Visual Quality:** It is common in video generation to evaluate the visual quality of videos, regardless of the semantic content. To evaluate visual quality, we use the Learned Perceptual Image Patch Similarity (LPIPS) (Zhang et al., 2018) (lower is better) over predicted and GT videos. For the Smth dataset, since videos contain single actions (meaning the AG contains a single triplet), we can report the Inception Score (IS) Salimans et al. (2016) (higher is better) and Fréchet Inception Distance (FID) (Heusel et al., 2017) (lower is better) using a TSM (Lin et al., 2019) model, pretrained on Smth. We note that we cannot report FID and IS on CATER since it provides multiple activities simultaneously, and hence does not support a pretrained video classifier. Finally, we also evaluate relative visual quality of two models by asking human annotators to select the video with higher quality. **b) Semantic Accuracy**: The key goal of AG2Vid is to generate videos which contain specified actions. To evaluate this, we ask human annotators to select which of two video generation

Table 2: Visual quality metrics of conditional video-generation methods in CATER and *Smth*. All methods use resolution $256 \times 256$ except for HG, which only supports $64 \times 64$.

| Resolution | Methods | Inception ↑ | FID ↓ | LPIPS ↓ | |
|---|---|---|---|---|---|
| | | Smth | Smth | CATER | Smth |
| 64x64 | Real Videos | $3.9 \pm 0.12$ | $0.0 \pm 0.0$ | $0.0 \pm 0.0$ | $0.0 \pm 0.0$ |
| | HG (Nawhal et al., 2020a) | $1.66 \pm 0.03$ | $35.18 \pm 3.6$ | − | $0.33 \pm 0.08$ |
| | **AG2Vid** (Ours) | $\mathbf{2.51 \pm 0.08}$ | $\mathbf{26.05 \pm 0.73}$ | $\mathbf{0.04 \pm 0.01}$ | $\mathbf{0.13 \pm 0.01}$ |
| 256x256 | Real Videos | $7.58 \pm 0.2$ | $0.0 \pm 0.0$ | $0.0 \pm 0.0$ | $0.0 \pm 0.0$ |
| | CVP (Ye et al., 2019) | $1.92 \pm 0.03$ | $67.77 \pm 1.43$ | $0.24 \pm 0.04$ | $0.55 \pm 0.08$ |
| | RNN | $1.99 \pm 0.05$ | $74.17 \pm 1.54$ | $0.14 \pm 0.05$ | $0.26 \pm 0.08$ |
| | V2V (Wang et al., 2018a) | $2.22 \pm 0.07$ | $67.51 \pm 1.42$ | $0.11 \pm 0.02$ | $0.29 \pm 0.09$ |
| | **AG2Vid** (Ours) | $\mathbf{3.02 \pm 0.11}$ | $\mathbf{66.7 \pm 1.29}$ | $\mathbf{0.07 \pm 0.02}$ | $\mathbf{0.25 \pm 0.08}$ |
| | **AG2Vid, GTL** (Ours) | $\mathbf{3.52 \pm 0.14}$ | $\mathbf{65.04 \pm 1.25}$ | $\mathbf{0.06 \pm 0.02}$ | $\mathbf{0.22 \pm 0.09}$ |

Table 3: Ablation experiment for components of the frame generation. Losses are added one by one.

| Loss | Inception ↑ | FID ↓ | LPIPS ↓ | |
|---|---|---|---|---|
| | Smth | Smth | CATER | Smth |
| Flow | $1.59 \pm 0.02$ | $107.26 \pm 1.46$ | $.14 \pm .01$ | $.70 \pm .06$ |
| + Perceptual | $2.21 \pm 0.07$ | $71.70 \pm 1.46$ | $.08 \pm .03$ | $.29 \pm .07$ |
| + Action Disc. | $\mathbf{3.02 \pm 0.11}$ | $\mathbf{66.7 \pm 1.29}$ | $\mathbf{.07 \pm .02}$ | $\mathbf{.25 \pm .08}$ |

models provides a better depiction of actions in the real video. The protocol is similar to the visual quality evaluation above. We also evaluated action timing, see below.

**Compared methods.** Generating videos based on action-graphs is a new task. There are no off-the-shelf models that can be used for direct evaluation with our approach, since no existing models take as input an action graph and output a video. To provide fair evaluation, we compare with two types of baselines. (1) First, existing baseline models that share some functionality with AG2Vid. (2) Second, variants of the AG2Vid model that shed light on its design choices. Each baseline serves to evaluate specific aspects of the model, as described next. **Baselines: (1) HOI-GAN (HG) (Nawhal et al., 2020b)** generates videos given a single action-object pair, an initial frame and a layout. It can be viewed as operating on a two-node action graph without timing information. we compare HG to AG2Vid on the Smth dataset because it contains exactly such action graphs. HG is not applicable to CATER data. **(2) CVP (Ye et al., 2019)** uses as input the first image and layout for future frame prediction without access to action information. CVP allows us to asses the visual quality of AG2Vid videos. However, it is not expected that CVP captures the semantics of the action-graph, unless the first frame and action are highly correlated (e.g., a hand at the top-left corner always moves downwards). **(3) V2V (Wang et al., 2018a)**: This baseline uses a state-of-the-art Vid2Vid model based on (Wang et al., 2018a) to generate videos from *ground-truth* layout. Since it uses ground-truth layout it provides an upper bound on Vid2Vid performance for this task. We note that Vid2Vid cannot use the action graph, and thus it is not provided as input. **AG2Vid variants: (4) RNN**: This AG2Vid variant replaces the layout generation GCN with an RNN that processes the action graphs. The frame generation part is the same as AG2Vid. More details are provided in the Supp Sec.4.1. **(5) AG2Vid, GTL:** An AG2Vid model that uses ground truth layout at inference time. It allows us to test if using the GT layout for all frames improves overall AG2Vid video quality and semantics.

**Layout Generation Ablations.** We experiment with an RNN architecture as an alternative to the GCN implementation of the LGF. The motivation behind the RNN experiment is to compare the design choice of the GNN to a model that processes edges sequentially (RNN). This RNN has access to the same input and supervision to the GCN, namely, $l_{t-1}$ and $A_t$, and the results from Table 4 confirm the advantage of GCN processing. For more details, see Sec. 4.1 in the Supplementary.

**Loss Ablations.** Table 3 reports ablations over the losses of FGF, confirming that the perceptual loss and actions discriminator losses improve the overall visual quality on CATER and Smth.

**Semantic and Visual Quality.** Fig. 4 shows sample videos generated by AG2Vid, and Fig. 5 shows comparison to generation by baselines. Table 1 shows the results of human evaluations for semantic and visual quality. It can be seen that AG2Vid is more semantically accurate and has better visual

quality than the baselines, and it is comparable to AG2Vid,GTL. See Sec 4.4 in Suppl for additional evaluations of AG2Vid correctness of the generated actions. Table 2 evaluates visual quality using several metrics, with similar takeaways as in Table 1.

**Action Timings.** To evaluate the extent to which the AG2Vid model can control the timing of actions, we generated AGs of actions at different times and asked annotators to choose in which video the action is executed first. In 89.45% of the cases, the annotators confirmed the intended result. For more information see Sec. 4.3 in Suppl.

**Composing New Actions.** To evaluate the extent to which the AG2Vid model can generalize at test time to unseen actions, we manually defined four compositions of learned actions. As seen in Fig. 6, we are using learned atomic actions to generate new action combinations that did not appear in the training data (either by having the same object perform multiple objects at the same time, or multiple objects performing coordinated actions). For example, in CATER, we created the actions "swap" based on "pick-place" and "slide" and "huddle" based on "contain". For Smth we composed the "push-left" and "move-down" to form the "left-down" action. For each generated video, raters were asked to choose the correct action class from a list of possible actions. The avg. class recall for CATER and Smth is $96.65$ and $87.5$ respectively, see the Supplementary for results by action.

## 6    DISCUSSION

We present a video-synthesis approach with a new Action Graph formalism, that describes how multiple objects interact in a scene over time. By using this formalism, we can synthesize complicated compositional videos and construct novel actions and action combinations. Although our approach outperforms previous methods, our model still fails in several situations. First, our model depends on the initial frame and layout. This could be potentially addressed by using an off-the-shelf image generation model. The formal AG representation is designed for describing complex semantic information in an easy-to-grasp way. The formalism could be further extended to handle other actions or their properties that were not present in today's datasets. For instance, it may be desired to capture features of actions described by adverbs. This can be achieved by adding attributes over actions, which we leave for future work. Finally, in this work we present an hierarchical and modular pipeline of video synthesis: first actions are scheduled for a specific timestep, then the scene layout is predicted, and finally the future flow is predicted and refined. While this pipeline is fairly general, we believe these representations can be further adopted to different datasets. For example, pose representation can be added for videos of people.

## 7    BROADER IMPACT

The paper proposes a new framework for video generation, which focuses on coordinated multiple simple actions, operating on simple daily objects. We believe it has potential for improving the quality and versatility of video generation. Video synthesis technology has many practical implications, such as generating simulated data for training robots and improving content search in video. These clearly have positive societal impact. The current work does not focus on generating faces or human movement, and as a result, we estimate that the potential for negative societal and ethic aspects is low.

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

In this supplementary file we provide additional information about our model, training losses, experimental results, and qualitative examples.

# 1 GRAPH CONVOLUTION NETWORK

As explained in the main paper, we used a Graph Convolution Network (GCN) (Kipf & Welling, 2016) to predict the layout $\ell_t$ at time step $t$. The GCN uses the structure of the action graph, and propagates information along this graph (in $K$ iterations) to obtain a set of layout coordinates per object.

Each object category $c \in \mathcal{C}$ is assigned a learned embedding $\phi_c \in \mathbb{R}^D$ and each action $a \in \mathcal{R}$ is assigned a learned embedding $\psi_a \in \mathbb{R}^D$. We next explain how to obtain the layouts $\ell_t$ using a GCN. Consider the action graph $A_t$ at time $t$ with the corresponding clocked edges $(i, a, j, r)$. Denote the layout for node $i$ at time $t - 1$ by $\ell_{t-1,i}$. The GCN iteratively calculates a representation for each object and each action in the graph. Let $\boldsymbol{z}_{i,k} \in \mathbb{R}^d$ be the representation of the $i^{th}$ object in the $k^{th}$ layer of the GCN. Similarly, for each edge in $A_t$ given by $e = (i, a, j, r)$ let $\boldsymbol{u}_{e,k} \in \mathbb{R}^d$ be the representation of this edge in the $k^{th}$ layer. These representations are calculated as follows. At the GCN input, we set the representation for node $i$ to be: $\boldsymbol{z}_{i,0} = [\boldsymbol{\phi}_{o(i)}, \ell_{t-1,i}]$. And, for each edge $e = (i, a, j, r)$ set $\boldsymbol{u}_{e,0} = [\boldsymbol{\psi}_a, r, \ell_{t-1,i}, \ell_{t-1,j}]$. All representations at time 0 are transformed to $D$ dimensional vectors using an MLP. Next, we use three functions (MLPs) $F_s, F_a, F_o$, each from $\mathbb{R}^D \times \mathbb{R}^D \times \mathbb{R}^D$ to $\mathbb{R}^D$. These can be thought of as processing three vectors on an edge (the subject, action and object representations) and returning three new representations. Given these functions, the updated object representation is the average of all edges incident on $i$:[5]

$$\boldsymbol{z}_{i,k+1} = \sum_{e=(i,a,j,r)} F_s(\boldsymbol{z}_{i,k}, \boldsymbol{u}_{e,k}, \boldsymbol{z}_{j,k}) + \sum_{e=(j,a,i,r)} F_o(\boldsymbol{z}_{j,k}, \boldsymbol{u}_{e,k}, \boldsymbol{z}_{i,k}) \tag{2}$$

Similarly, the representation for edge $e$ is updated via: $\boldsymbol{u}_{e,k+1} = F_a(\boldsymbol{z}_{i,k+1}, \boldsymbol{u}_{e,k}, \boldsymbol{z}_{j,k+1})$.

Finally, we transform the GCN representations above at each time-step $t$ to a layout $\ell_t$ as follows. Let $K$ denote the number of GCN updates. The layout coordinates of $\ell_{t,i}$ are the output of an MLP applied to $\boldsymbol{z}_{i,K}^t$, which are simply the set of the predicted normalized bounding box coordinates. The object descriptor is $\boldsymbol{z}_{i,K}^t$.

# 2 LOSSES AND TRAINING

We elaborate on the Flow and Perceptual losses from Section 4.2.

**Optical flow loss $\mathcal{L}_F$.** The flow loss $\mathcal{L}_F$ is the warping loss which measures the error between the warps of the previous frame and the ground truth of the next frame $v_t^{GT}$.

$$\mathcal{L}_f = \frac{1}{T-1} \sum_{t=1}^{T-1} \|w_t - v_t^{GT}\|_1 \tag{3}$$

where $w_t = W(f_t, v_{t-1})$ as defined in Section 4.1. This flow loss proposed previously in Wang et al. (2018a); Zhou et al. (2016).

**Perceptual loss $\mathcal{L}_P$.** This is the standard perceptual loss as in pix2pixHD (Wang et al., 2018b). In particular, we use the VGG network (Simonyan & Zisserman, 2014) as a feature extractor and minimize the error between the extracted features from the generated and ground truth images from $L$ layers.

$$\mathcal{L}_P = \sum_l^L \frac{1}{P_l} ||\phi^{(l)}(v_t) - \phi^{(l)}(v_t^{GT})||_1 \tag{4}$$

---

[5]Note that a box can appear both as a "subject" and an "object" thus two different sums in the denominator.

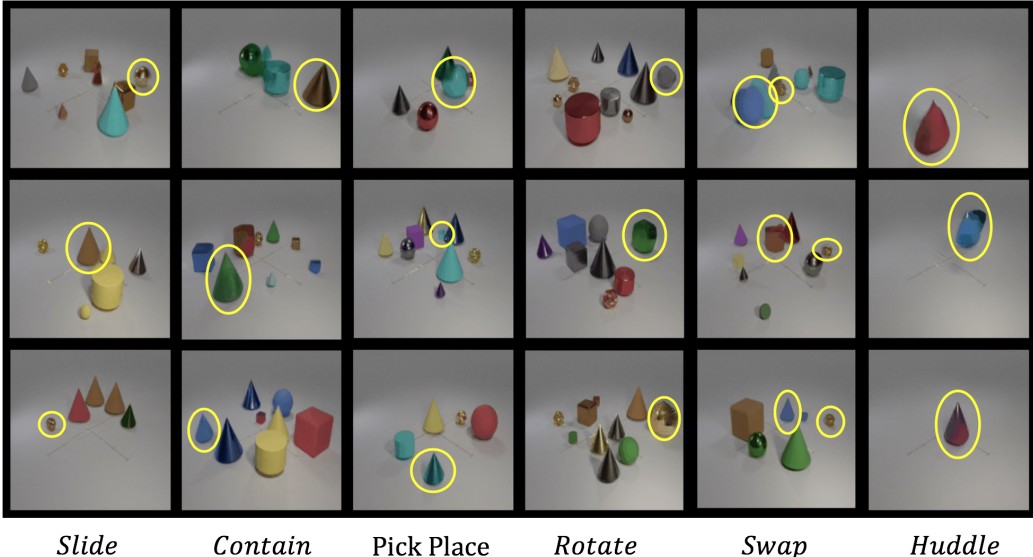

| *Slide* | *Contain* | Pick Place | *Rotate* | *Swap* | *Huddle* |

Figure 7: Qualitative examples for the generation of actions on the CATER dataset. We use the AG2Vid model to generate videos of four standard actions and two composed unseen actions ("Swap" and "Huddle"). The objects involved in actions are highlighted. *Click the image to play the video clip in a browser.*

where $\phi^{(l)}$ denotes the $l$-th layer with $P_l$ elements of the VGG network. We sum the above over all frames in the videos.

The overall optimization problem is to minimize the weighted sum of the losses:

$$\min_\theta \max_{D_A} \mathcal{L}_A(D_A) + \lambda_\ell \mathcal{L}_\ell + \lambda_f \mathcal{L}_f + \lambda_P \mathcal{L}_P \,, \tag{5}$$

where $\theta$ are all the trainable parameters of the generative model, $\mathcal{L}_\ell$ is the Layout loss, and $\mathcal{L}_A$ is the pixel action discriminator loss from Section 4.2. In addition to the loss terms in Equation 5, we use a feature matching loss (Larsen et al., 2016; Wang et al., 2018b) to match the statistics of features extracted by the GAN discriminators.

## 3 ACTIONS

For the *Something Something* dataset (Goyal et al., 2017), we use the eight most frequent actions. These include: "Putting [something] on a surface", "Moving [something] up", "Pushing [something] from left to right", "Moving [something] down", "Pushing [something] from right to left", "Covering [something] with [something]", "Uncovering [something]", "Taking [one of many similar things on the table]" . See Figure 8 for qualitative examples. The box annotations of the objects from the videos are taken from Materzynska et al. (2020).

| Model | mIOU ↑ | | R@0.3 ↑ | | R@0.5 ↑ | |
|---|---|---|---|---|---|---|
| | CATER | Smth | CATER | Smth | CATER | Smth |
| Random | 5.05 | 13.55 | 5.94 | 16.50 | 01.86 | 4.81 |
| RNN | 75.71 | 41.28 | 80.67 | 61.70 | 78.91 | 39.23 |
| AG2Vid | **93.09** | **51.32** | **99.55** | **74.50** | **98.04** | **53.85** |

Table 4: Layout generation evaluation.

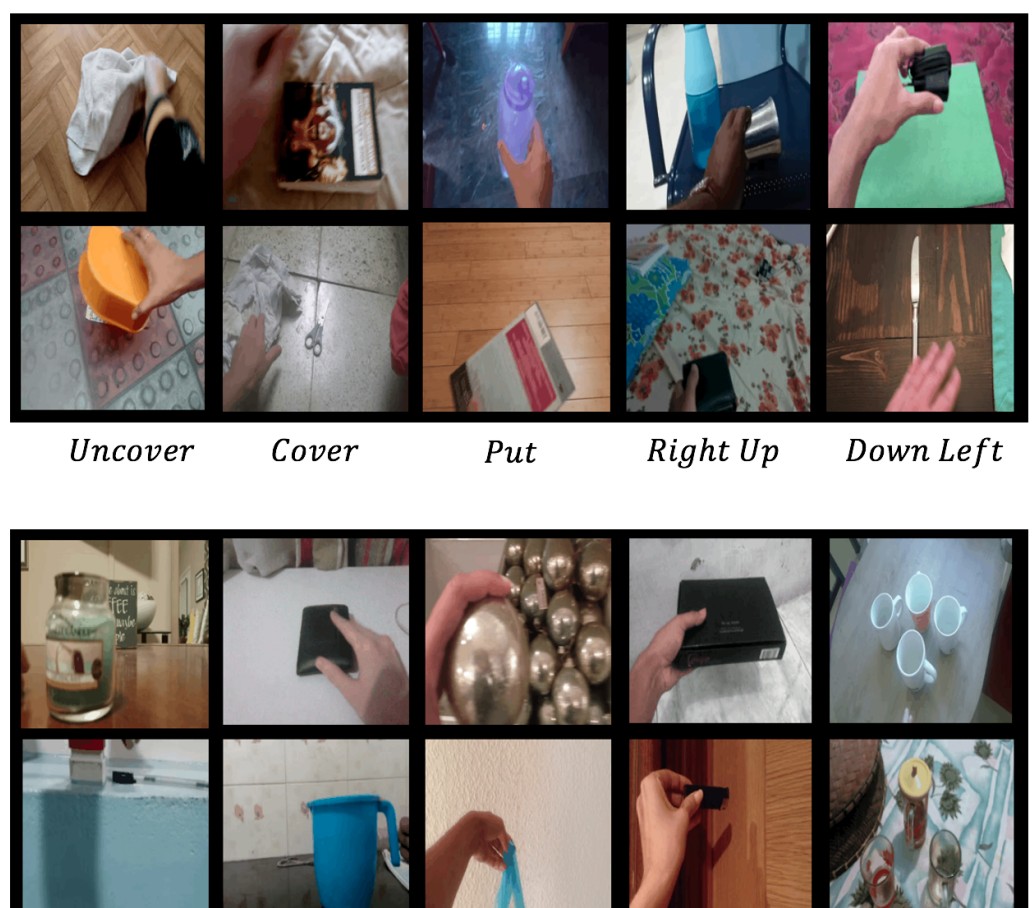

Figure 8: Qualitative examples for the generation of actions on the Something Something dataset. We use our AG2Vid model to generate videos of eight standard actions and two composed unseen actions ("Right Up" and "Down Left"). *Click the image to play the video clip in a browser.*

## 4 EXPERIMENTS AND RESULTS

### 4.1 RNN BASELINE

We experiment with an RNN architecture as an alternative to the GCN implementation of the LGF. This RNN has access to the same input and supervision to the GCN, namely, to $l_{t-1}$ and $A_t$. We next explain how to obtain the layouts $\ell_t$ using the RNN.

Each object category $c \in \mathcal{C}$ is assigned a learned embedding $\phi_c \in \mathbb{R}^D$ and each action $a \in \mathcal{R}$ is assigned a learned embedding $\psi_a \in \mathbb{R}^D + 1$. Consider the action graph $A_t$ at time $t$ with the corresponding clocked edges $(i, a, j, r)$.

let $U_i \in \mathbb{R}^{|E| \times 4D+1}$ denote the matrix, where every row that corresponds to edge is comprised of the object, action, subject embeddings, the embedding of the $i$th object, and the target the progress feature. We apply the RNN over $U_i$ and denote $z_{i,t}$ as the last hidden state of the result. The new object descriptor of $l_{t,i}$ is then $z_{i,t}$, and to obtain a new bounding box location, an MLP is applied over $z_{i,t}$. $l_t$ is the new updated bboxes and object descriptors. The RNN model has 3 layers and 512 hidden layer size.

| Composed Actions | | | |
|---|---|---|---|
| Swap | Huddle | RU | DL |
| 92.1 | 98.6 | 75.0 | 100. |

| | Standard Actions | | | Composed Actions | |
|---|---|---|---|---|---|
| Slide | Contain | Pick Place | Rotate | Swap | Huddle |
| 96.7 | 100.0 | 90.0 | 56.7 | 93.3 | 100.0 |

Table 5: Human evaluation of the **semantic accuracy** of the actions in the generated videos.

Table 6: Human evaluation of **timing** in generated videos (see section 4.3). The table reports accuracy of human annotator answer with respect to the true answer.

## 4.2 LAYOUT ACCURACY

AG2Vid produces bounding boxes of objects as a function of time. Since our datasets contain ground-truth boxes, we can compare our predictions to these. We evaluate the intersection over union (IOU) over the predicted and ground truth boxes. We report the mean intersection over union (mIOU) which is the mean over the entire set of boxes. Additionally, we measure the recall over the by considering an object to be a correct detected if the IOU between the GT and predicted box is higher than 0.3 (R@0.3) or 0.5 (R@0.5). Results are reported in Table 4. It can be seen that the RNN underperforms, supporting the choice of GCN for layout generation in the AG2Vid model. The RNN is likely to under-perform as it assumes order over the AG list of edges, which is not a natural way to process a graph.

## 4.3 HUMAN EVALUATION OF ACTION TIMING IN GENERATED VIDEOS

As described in Section 5.1, we evaluated to which extent the action graphs (AGs) can control the timing execution of actions on the CATER dataset. Thus, we generated 90 pairs of action graphs where the only difference between the two graphs is the timing of one action. We then asked the annotators to select the video where the action is executed first. The full results are depicted in Table 6, and visual examples are shown in Figure 9. The results for all actions but "Rotate" are consistent with the expected behavior, indicating that the model correctly executes actions in a timely fashion. The "Rotate" action is especially challenging to generate since it occurs within the intermediate layout. It is also easier to miss as it involves a relatively subtle change in the video.

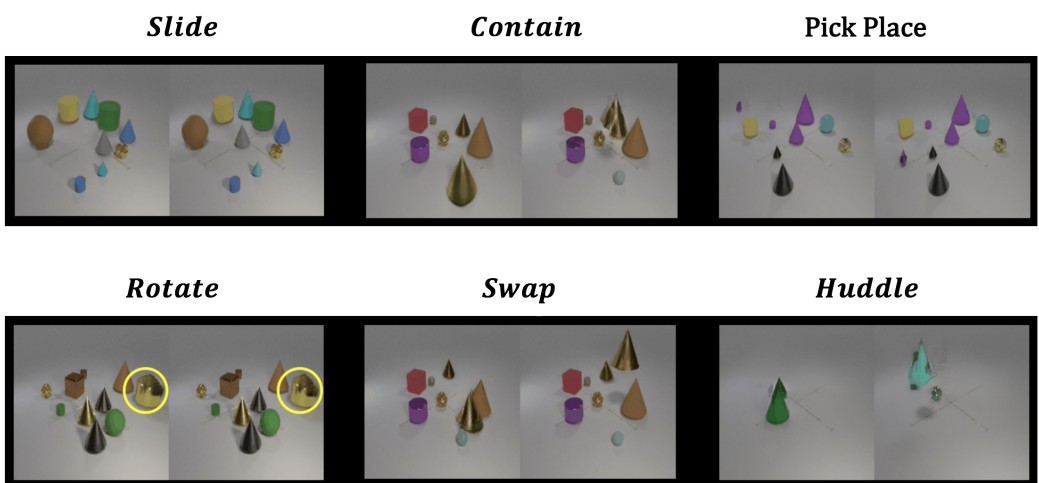

Figure 9: Timing experiment examples in CATER. We show the clock edges can manipulate the timing of a video by controlling when the action is performed to achieve goal-oriented video synthesis. The objects involved in "rotate" are highlighted. *Click the image to play the video clip in a browser.*

## 4.4 HUMAN EVALUATION OF SEMANTIC QUALITY IN GENERATED VIDEOS

To test the degree to which the generated videos match their corresponding actions, we generated twenty videos per action for the Something-Something dataset and asked three different human annotators to evaluate each video. Each annotator was asked to pick the action that best describes

the video out of the list of possible actions. We provide the results in Fig. 7. Each cell in the table corresponds to the class recall of a specific action. To determine if a video correctly matches its corresponding action, we used the majority voting over the answers of all annotators.

It turns out that humans do not perform perfectly in the above task. We quantified this effect in the following experiments on the Something-Something dataset. We used the above annotation process for ground-truth videos (see "Real" row in Table 7). Interestingly, it can be seen from the reported accuracy in Table 7 that our generated action videos of "Move Down" and "Take" are more easily recognizable by humans than the ground truth videos. For the CATER dataset, we did not perform such human evaluation of predicted actions, since CATER videos contain multiple activities.

To evaluate the extent to which the AG2Vid model can generalize at test time to unseen actions, we manually defined four compositions of learned actions. In Table 5, we show the semantic accuracy of the human evaluation we did for the new unseen actions: "huddle", "Swap", "Right Up" and "Down Left".

| Video Source | Standard Actions | | | | | | | |
|:---:|:---:|:---:|:---:|:---:|:---:|:---:|:---:|:---:|
| | Right | Up | Down | Left | Put | Take | Uncover | Cover |
| Generated | 100. | 50. | 100. | 75. | 95. | 80. | 25. | 55. |
| Real | 100. | 100. | 90. | 100. | 100. | 65. | 100. | 85. |

Table 7: The semantic quality evaluated by humans of the generated and real action videos. We asked raters to select the action described in the video for each synthesized video with a given action. The table reports the accuracy of the human annotators with respect to the true action underlying the video. Actions above correspond to: 'Pushing [something] from left to right', 'Moving [something] up', 'Moving [something] down', 'Pushing [something] from right to left', 'Putting [something] on a surface', 'Taking [one of many similar things on the table]', 'Uncovering [something]', 'Covering [something] with [something]' .

## 4.5 COMPARING AG2VID TO SCENE-GRAPH BASED GENERATION

Scene graphs are an expressive formalism for describing image content. Both datasets we use have frame-level scene graph annotation. Thus, we wanted to compare generation from these scene graphs with generation from action graphs. Towards this end, we used a scene-graph-to-image model (Johnson et al., 2018) trained to generate the images in the videos from their corresponding scene graphs. This model does not condition the action or initial frame and serves only for comparison in terms of realistic generation. It can be seen in Figure 10 that the temporal coherency of AG2Vid is more consistent and coherent than the sequence of scene graphs.

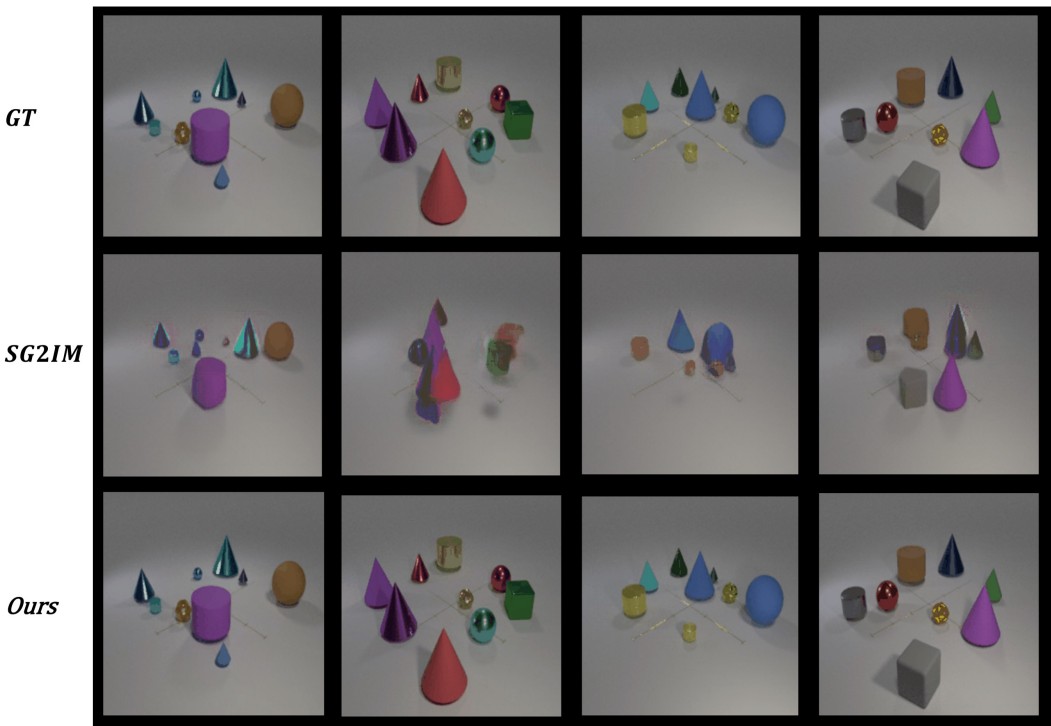

Figure 10: Comparing Sg2Im and Ag2Vid results in CATER. Each column is a different sample. *Click the image to play the video clip in a browser.*

