# OpenReview forum: "Compositional Video Synthesis with Action Graphs"
_ICLR.cc/2021/Conference — Reject_

### Official Review · AnonReviewer2 · 2020-10-26
**Sound method for video generation based on action graphs**

**Rating:** 7
**Confidence:** 3

**Review:**

This paper proposes a generative method (AG2Vid) that generates video conditioned by the first frame, first layout and an action graph. An action graph is defined such that nodes represent objects in the scene and edges represent actions. To capture the temporal dynamics, each pairwise connection is enriched with a time interval to indicate the temporal segment when the action happens. For each time step, the method consists of several stages: First, it creates the layout corresponding to the current time step based on the current graph and previous layout. Then it extracts the optical flow based on the last two layouts and the previous generated frame and finally, it generates the current frame (at pixel level) based on the predicted optical flow and the previous frame. Several metrics, including human evaluation, indicates that the method outperforms powerful baselines on two datasets: CATER and Something-Something v2.

Pro:
- Generating video content is a difficult task and the idea of generating frames based on action graphs to more explicitly focus on the activity class is interesting and naturally integrated into the proposed architecture.
- The method clearly outperforms the baselines and produces high-quality videos.
- The experiments regarding the generalization to novel compositions of actions are interesting and show promising results for generating videos beyond the training domain.
- The paper is generally well written, with clear explanations on the main aspects and a good balance between quantitative evaluation and qualitative examples.

Cons:
- Since Action Genome [1] dataset provides more complex scene-graph annotations for videos in Charades, quite similar to the one required in this work (especially the contact subset of Action Genome), why do the authors choose Something-Something dataset instead of Charades?
- From the paper, it seems to me that the subset of videos picked from Smt-Smt dataset only contains 2 objects (nodes), thus the action graph is very simple and the temporal segment covers the entire video. This aspect is only briefly mentioned in the main paper. Moreover, I think the ability of the method would be more clearly demonstrated in classes that have more than 2 objects if the extraction of the AG would be possible in that case.
- The RNN ablation is interesting, showing the necessity of GNN processing. However, more details about the motivation and intuition behind these experiments should be added in section 5.

Minor:
- I think there is a typo in Eq (4). Shouldn’t VGG be applied also on the predicted v_t?
- In the same manner, as the compositional experiment, it would be interesting to test the model using the same first frame from training videos, but changing the action labels from the Action Graphs (on Smt-Smt). In this way, it would be clearer that the model doesn’t use any kind of biases in the dataset.

Since video generation could be a sensitive task, especially when conditioned on a set of actions, the ethical aspect of that work should be taken into consideration and discussed.

[1] Action Genome: Actions as Compositions of Spatio-temporal Scene Graphs, Ji et. al, CVPR 2020

I found the proposed method interesting and suitable for the video generation tasks. Moreover, both the ablation study and the quantitative evaluation show good performance, so I recommend the acceptance.

########### UPDATE #########

I thank the authors for their responses and for updating the paper. I think this work introduces some new and valuable ideas for generating videos conditioned by an action graph and I recommend the acceptance.

---

> ### Author Response · Authors · 2020-11-20
> **Response to R2**
>
> Thank you for the constructive comments and suggestions. We address your comments below.
>
> **Q1: Action Genome dataset provides more complex scene-graph annotations for videos in Charades. Why do the authors choose the Smth dataset instead of Charades?.**
> The Action Genome dataset is similar to Smth, though considerably smaller (Action Genome contains 10K videos, compared to 200K for Smth). We chose to focus on Smth since Action Genome was only very recently released. We note that our approach does not require annotated scene graphs as provided in Action Genome, but could be extended to use those if available. We consider this an interesting extension to explore. We revised the Related Work according to your comment.
>
> **Q2: It seems to me that the subset of videos picked from Smt-Smt dataset only contains 2 objects (nodes), thus the action graph is very simple and the temporal segment covers the entire video?.**
> We considered the eight most frequent actions in Smth (e.g., “Putting [something] on a surface”, “Covering [something] with [something]”). Restricting the dataset to these resulted in videos that contain up to three objects (resulting in an AG with three nodes), including the “Hand” object which appears in most videos. For example, in the action “Putting [something] on a surface”, there is the “Hand” node and an additional object node (e.g. a cup). Similarly, in the “Covering [something] with [something]” there is the “Hand” and two more objects. Regarding temporal segments, indeed for training we took the time of the action to be the entire video. We note that learning from this data still allowed us to generate videos of new action compositions.
>
> **Q3: I think the ability of the method would be more clearly demonstrated in classes that have more than 2 objects if the extraction of the AG would be possible in that case.**
> We agree that the multi-object setting is the most interesting one. For CATER, we do demonstrate generation in this setting. For SMTH, we are restricted by the structure of the dataset but note that we do have a three object setting even there.
>
> **Q4: The RNN ablation is interesting, showing the necessity of GNN processing. However, more details about the motivation and intuition behind these experiments should be added in section 5.**
> We agree that the motivation is missing, which is an unfortunate consequence of the page limit. To predict the layout of the objects given an AG at time t, we want to propagate the actions and objects information across the different object representations. A natural way to do this is using a GNN. However, we also explore an RNN model that produces new object representations by sequentially processing the edges of the AG. We revised the manuscript in the Experiments section to include this.
>
> **Q5: Minor - typo in Eq (4). Shouldn’t VGG be applied also on the predicted v_t?** Thank you for noticing. We defined in Section 2 in the supplementary, $\phi(l)$ as the $l$-th layer with $P_l$ elements of the **VGG** network, thus the VGG is redundant: $||\phi^{(l)}(v_{t})-\phi^{(l)}(v^{GT}_{t})||_1$. We revised the equation accordingly.
>
> **Q6: Minor - It would be interesting to test the model using the same first frame from training videos, but changing the action labels from the Action Graphs (on Smt-Smt).**
> Thank you for this suggestion. To provide more evidence that our model indeed relies on the AG, we include more synthesized videos [1] of four different actions: "Pushing [something] from right to left",  "Pushing [something] from left to right", "Moving [something] up" and "Moving [something] down". We chose these specific actions as they are most straightforward to verify.
>
> [1] * https://youtu.be/STqI4s_Akd4 (for best quality, choose settings -> quality -> 1080p )
>
> **Q7: Ethical - Since video generation could be a sensitive task, especially when conditioned on a set of actions, the ethical aspect of that work should be taken into consideration and discussed.** As with any generative approach, we agree that one should be aware of the potential misuse of the technology. Content generation, including images and videos, has many beneficial applications from sim-to-real for training agents, through reducing cost and "democratizing" content generation in entertainment and gaming, to improving educational and instructional videos.

---

> > ### Comment · AnonReviewer2 · 2020-11-23
> > **RE: Response to R2**
> >
> > Thank you for the careful response and for addressing my concerns in the revised version of the paper. I agree with the argument about the difference of size between Something-Something dataset and Action Genome.
> >
> > However, I have some minor observations regarding the additional paragraph in the related work. To me, Something-Something-v2 could not be considered as “it includes daily actions collected from the web” since the dataset “was created by a large number of crowd workers” (https://20bn.com/datasets/something-something).
> >
> > Moreover, since there is no comparison between SG and Action Graph in terms of performance, I feel that the statement “[we] argue [Action Graph] is more natural for representing videos of actions” is a little bit too strong.

---

> > > ### Author Response · Authors · 2020-11-24
> > > **RE: RE: Response to R2**
> > >
> > > **Q: “Something-Something-v2 could not be considered as “it includes daily actions collected from the web” since the dataset “was created by a large number of crowd workers”**
> > > Thank you for the comment. The use of “collected from the web” was meant to emphasize the diversity of Something-Something-V2. However, we agree that this is not precise and revise the manuscript accordingly. Generally, we agree that Action Genome is an exciting new resource and will explore using it in future work.
> > >
> > >
> > >
> > > **Q: “there is no comparison between SG and Action Graph in terms of performance, I feel that the statement “[we] argue [Action Graph] is more natural for representing videos of actions” is a little bit too strong.”**
> > > Thank you for this comment. We agree that the statement is a bit too strong and have removed it in the latest revision.

---

### Official Review · AnonReviewer4 · 2020-10-27
**Interesting method, some issues regarding the novelty and training settings.**

**Rating:** 6
**Confidence:** 4

**Review:**

This paper proposes a model for video generation which disentangles the object layout prediction, frame-by-frame, from the actual pixelwise frame generation. A so-called Action Graph (AG) is used as specification of the video to be generated, rather than a sentence. Action graphs model objects as nodes and actions as clocked edges. This way action graphs are "clocked" so to take into account the current progress of each action. A Graph Convolutional Network (GCN) is used to process the action graph and predict the next layout. The GCN is fed with the previous layout and the current AG. The final frame is generated by warping the previous frame and predicting an additive signal to the warped output. The network is trained similarly to a GAN.
Experiments on two datasets are provided human and quantitative evaluation show superior results wrt to the baseline

Strengths
- well motivated approach for video generation.
- experimental results show better performance with respect to segmentation based video generation


Weaknesses

- one main limitation of the method is the lack of a specific procedure to obtain action graphs. This reflects both at training time and at inference time but it is more critical at training time. The level of detail of annotation required is high and fine grained. How would this work to obtain action annotations for a dataset such as Kinetics, AVA or EPIC-KITCHENS?
-unclear use of GAN training. GAN-like training is exploited but it is not clear why this is necessary. Moreover, apparently there is  no noise injection allowing to generate multiple videos from the same action graph. What would happen if the loss in Eq (1) is dropped in favor of only the perceptual losses? Is the GAN loss required to impose "veracity" on the layout generation? Given that GT layout are available why not adding some layout consistency loss? This could be done by optimizing the IoUs of objects [a].
- How do you manage similar AG corresponding to different output frames? Is the GAN loss used for this? What happens if you remove it?


Novelty and related work.

Why Nawhal et al., ECCV 2020 is not discussed in the related work section? The proposed approach seems largely inspired by it and HOI-GAN is even used as a baseline, a fair writing of the related work and introduction should mention and discuss in detail how this work improves over it (are the timed edges the main addition?).
The concept of action progress has been used before with a very similar definition to the one used here.




Regarding the model:
Layout are sets of bounding box coordinages + "features". The model demands to the "feature" to encode all information which is not included in the box coordinates such as pose, lighting, texture, color etc.
Some of these variables could be modelled explicitely such as 3D or 2D pose. Why not encoding the object pose in the layout? The pose of the object could be modelled as a latent variable if not available as annotation and yet influence the optical flow generation.

Regarding the experiments:
HOI-GAN, Nawhal et al. 2020, has been tested on EPIC-KITCHENS which is a more complex real-world setting wrt to Something-Something and CATER can AG2Vid be applied in such context? if not why?

A major issue of this work lies in the novelty with respect to HOI-GAN, which is used as a baseline but not discussed in the related work section, plus the comments above make the paper a nice contribution but just marginally above the threshold.

References

[a] PolarMask: Single Shot Instance Segmentation with Polar Representation, 2020
[b] UnitBox: An Advanced Object Detection Network,2016
[c]  Am I Done? Predicting Action Progress in Videos, 2017
[d] Temporal Cycle-Consistency Learning, 2019

---

> ### Author Response · Authors · 2020-11-20
> **Response to R4**
>
> Thank you for the constructive comments and suggestions. We address your comments below.
>
> **Q1: HOI-GAN is not discussed in RW. A major issue of this work lies in the novelty with respect to HOI-GAN.** HOI-GAN is clearly related work, and in our experiments we compare to it. We also had it in the related work, but the paragraph was accidentally left out in the submitted version. We added it in the current version. The focus of AG2VID is, however, very different from HOI-GAN. Our focus is on generation of multiple simultaneous actions over time, performed by multiple objects. Our modeling approach directly addresses this challenge via the notion of clocked edges. On the other hand, the goal in HOI-GAN is to generate a single action performed on a single object (e.g., lift fork), and it does not support the timing of actions. The AG2VID model allows us to create complex multi-action videos as in the CATER dataset, which cannot be done with HOI-GAN, and we can also create new actions as shown for CATER and SMTH, which HOI-GAN cannot do. We did think it was useful to compare to HOI-GAN in the setting of generating a single action, and in that setting AG2VID outperforms HOI-GAN  (see Table 2 and 3).
>
> **Q2: One main limitation of the method is the lack of a specific procedure to obtain action graphs. The level of detail of annotation required is high and fine grained. How would this work to obtain action annotations for a dataset such as Kinetics, AVA or EPIC-KITCHENS.**  One natural way to automatically obtain weak-AG annotations is by running a pre-trained action detector and an object detector model followed by relevant postprocessing. Another natural step is to consider ''latent action graphs'' that are learned in an unsupervised manner, but we leave this for future work.
>
> **Q3: HOI-GAN has been tested on EPIC-KITCHENS which is a more complex real-world setting wrt to Smth and CATER can AG2Vid be applied in such context? if not why?** The goal of our work was to propose a general framework for goal-oriented video synthesis of actions. Towards this end, we chose to use two datasets: CATER, which contains multiple actions (simultaneously), and Smth, which is a diverse and complex real-world dataset. Indeed EPIC-KITCHENS is another dataset we could apply AG2Vid to; however, we choose Smth for the following reasons. First, the Smth dataset is one of the most extensive datasets with almost ~200K videos of 174 actions and ~30K different objects (compared to ~40K instances, 125 actions, and 352 objects in EPIC-Kitchens). Additionally, it is also more diverse since it contains basic human activities created by a large number of crowd workers (not necessarily ego-centric kitchen videos) and without any limiting conditions such as fixed camera angles, similar appearance, or a small set of objects.
>
>
> **Q4: unclear use of GAN training. GAN-like training is exploited but it is not clear why this is necessary. Moreover, apparently there is no noise injection allowing to generate multiple videos from the same action graph. What would happen if the loss in Eq (1) is dropped in favor of only the perceptual losses? Is the GAN loss required to impose "veracity" on the layout generation?**
> Indeed, we train GANs and do not condition on noise, as motivated by previous works [1,2,3]. Specifically, previous works argue that there is enough variability in the input of the generator even in the absence of noise, and therefore it is not a necessity for GANs training. As for the effect of the GAN loss, removing it leads to a substantial drop in the visual quality as well as the quantitative results (see Table 3).
>
> **Q5: The concept of action progress has been used before with a very similar definition to the one used here.** Thank you for referring us to this work. We now added a relevant citation to the Related Work section. However, we note that while in [4] action progress served as a regression target, here we use the action progress as part of an execution mechanism ("Clocked Edges") to learn and control the timing for AGs, which we were the first to propose and utilize.
>
> **Q6: The model demands the "feature" to encode all information which is not included in the box coordinates such as pose, lighting, texture, color etc. Some of these variables could be modelled explicitly such as 3D or 2D pose. Why not encoding the object pose in the layout? The pose of the object could be modelled as a latent variable if not available as annotation and yet influence the optical flow generation.** This is a good idea, and we will explore adding it to the model to improve pose generation. We also added a discussion of this to the paper.
>
> [1] Deep multi-scale video prediction beyond mean square error, ICLR 2016.
>
> [2] Image-to-Image Translation with Conditional Adversarial Networks, CVPR 2017.
>
> [3] High-Resolution Image Synthesis and Semantic Manipulation with Conditional GANs, CVPR 2018.
>
> [4] Am I Done? Predicting Action Progress in Videos.

---

### Official Review · AnonReviewer1 · 2020-10-29
**Action Graphs Assist Video Generation.**

**Rating:** 5
**Confidence:** 2

**Review:**

This paper describes a method to generate videos from an initial image and a graph based description of how the scene should evolve.
The description is called "Action Graphs" and consists of a nodes representing objects, along with edges that describe actions between objects that should occur over given time intervals.
The method for producing a video is split into 4 parts:
1) An initial image frame and set of objects with bonding boxes are provided.
2) The action graph is "unrolled", and a graph is instantiated for each time step. The time intervals for actions are interpolated into 0->1 ranges and this is stored in the state of the edges for each time step. This part is algorithmic with no training.
The task of generating the video is then as follows..
3) The layout generating Function. This is a Graph Convolutional network on a graph consisting of the "unrolled" action graph at time t (and the graph at t-1?) and the previous positions of the objects. It's task is to predict positions of objects and their bounding boxes for the next frame.
4) The Frame generating Function. This takes the predicted layout, and the pixels from the previously generated frame and generates the next frame pixels. This uses previous techniques including a "flow prediction network".

Reason For Score:

This is a weak reject. The action graphs are a succinct way of declaring an evolving scene with multiple objects.
And the graph network used to interpolate object positions across a sequence of unrolled AGs is good. My main concern is for the utility of the methods outside the synthetic CATER dataset. And the contribution of the Action Graph to the "something something" dataset results is not clear.

Pros

The videos produced for the CATER dataset are nice, with objects in general retaining their boundary shapes and color/texture characteristics as they move. The movements themselves appear accurate and are well timed.
Most impressive are the compositional actions "eg. huddle and swap" in Fig6 which show ability to produce videos of unseen group actions (that are compositions of previously seen individual actions).

The work is quite thorough. They carry out visual quality assessments and ablation experiments.

It is possible that others in the community can rally around and develop the action graph description.
Scenario/Storyboard descriptions of evolving scenes are necessary for work on scene understanding, and there aren't any clear candidates to rally around.

Cons

What is the actual definition of the Action Graphs used?
For CATER it is stated that the target positions are provided for some objects, and elsewhere that an angle can be included for rotate.
Can we see the "formal" descriptions of nodes, edge types and edge attributes used?
Is it possible to have more than one action per object per time step? This seems possible in the novel "something something" examples, but I can't see how this would be implemented in the Action Graph for one object without 2 edges to the same object.

In the something-something dataset, are all the actions in this dataset single actions for one object with an edge from the single node to itself? If so, what is the contribution of the action graph?
What are the object(s) in "Putting (something) on a surface"?

The methods employed: graph neural network and optical flow based video generation not all that original, the contributions are in the Action Graph and the combination and application of known techniques (albeit seemingly well chosen for the CATER dataset).

The utility of Action Graphs is not clear beyond synthetic data, particularly the CATER dataset which has discrete action/movement time segments with well defined actions to occur within the segments. In the case of this paper, all the object identities and actions over discrete time steps are given directly to the graph model to do "scene graph interpolation". Another application would need a similar ground truth dataset for training, and it is not clear one can be made from real data.

---

> ### Author Response · Authors · 2020-11-20
> **Response to R1**
>
> Thank you for your constructive comments and suggestions. We address the comments below.
>
> **Q1: The utility of Action Graphs is not clear beyond synthetic data, particularly the CATER dataset.** In this work, we demonstrated the advantages of using AGs for synthesizing goal-oriented videos. For the Smth dataset (which is not synthetic), our approach can model more than one object whereas previous work [1] only handled up to a single object. This is the case for example, in the action, “Covering [something] with [something]”. We also showed that it is possible to generate new actions, composed of existing ones (Figure 6), and to control the timing of actions using AGs. These advantages are not limited to the CATER dataset and are also applicable to Smth and potentially other non-synthetic datasets. We do agree that AGs are a useful representation for modeling simultaneous and complex events like those in CATER. We hope that our work will encourage the community to invest in more complex datasets in the near future.
>
> **Q2: Another application would need a similar ground truth dataset for training, and it is not clear one can be made from real data.** One way to automatically obtain weak-AG annotations is by running a pre-trained action detector and an object detector model followed by relevant postprocessing. We agree that there is room for developing other procedures, which we leave for future work.
>
> **Q3: What is the actual definition of the Action Graphs used?. For CATER it is stated that the target positions are provided for some objects, and elsewhere that an angle can be included for rotate. Can we see the "formal" descriptions of nodes, edge and edge attributes used?**  In Something-Something, every object/action has a single associated object class with no attributes included. In CATER, every object has color, shape, material, and size attributes, which are the standard CATER attributes. Every CATER action is associated with a single action class (e.g., rotate). Specifically, “Pick Place” and “Slide” have destination coordinate attributes, which are the (x,y) coordinates of the target position. We included this information in Supplementary Material Section 3. We acknowledge this might be easy to miss and revise the main manuscript such that it includes Supplementary Material Section 3.
>
> **Q4: Is it possible to have more than one action per object per time step?.**
> Yes, the AG formulation allows for multiple simultaneous actions over the same object per timestep (see Section 3). Indeed, Figure 6 used this property to define the action “Right Up” by setting edges E to:
> E = {(Hand, Move Up, Cup, 0, 10), (Hand, Move Right, Phone, 0, 10)}
>
> **Q5: In the Smth dataset, are all the actions in this dataset single actions for one object with an edge from the single node to itself? If so, what is the contribution of the action graph? What are the object(s) in "Putting (something) on a surface?"**
> Not exactly. In every action in Smth, there is the “Hand” node (which is the hand performing the action) and at least one object which is involved in the action (See Figure 6, top). For example, in the action “Putting (something) on a surface,”  there are the “Hand” node and an additional object node (e.g., a cup), and thus the Action Graph contains {(Hand, Put, Cup, 0, 10)}. Similarly, in the “Covering [something] with [something],” there is the “Hand” and two more objects involved, such as {(Hand, Take, Towel, 0, 10), (Towel, Cover, Scissors, 0, 10)} (See “cover” action in Figure 4). As mentioned above, the AG modeling supports a variable number of objects compared to previous works [1]. Additionally, we show that it is possible to create new actions that are compositions of existing ones and control the timing of actions using the AG representation, which was not done before.
>
> [1] Generating Videos of Zero-Shot Compositions of Actions and Objects

---

### Official Review · AnonReviewer3 · 2020-10-29
**Reviewer #3**

**Rating:** 7
**Confidence:** 3

**Review:**

Overview:

The paper proposes a hierarchical approach to video synthesis based on Action Graph. Action Graph is a graph representation to describe the dynamics of individual objects. Based on this, the authors proposes an action scheduling mechanism to track the progress of action and then generate the scene layout at each timestamp. Finally, the pixels are generated based on the predicted scene layout. Experiments show that such AG2Vid paradigm can generate images on CATER and Something-Something dataset with a better quality compared to the baselines. It can also generate novel actions, treated by a composition of seen actions.

Strengths:

++ The idea of three-level abstraction (action schedule + scene layout generation + pixel generation) is sound. The formulation based on the idea is neat and straightforward.

++ The experiments on generating multiple actions, single action, as well as novel actions indicate that the method is capable of disentangling and composing atomic action for individual object.


Weaknesses:

-- In the stage of pixel generation, it seems that the frame generation function cannot handle the overlap case very well. For example, in the "pick place" video in Figure 4, when the green cone is picked up and moved towards, it should have occluded the yellow cone at certain steps. However, it is occluded **by** the yellow one instead in the generated video. But since the masks
$ M_{t-1}, M_t $ are given individually, I personally think it should not be the case: since there is only one object moving (i.e. green cone), you can warp that specific mask and simply stick that on top of the original frame. Can the authors explain why does it fail?

-- The results on Something-Something indicate that flow-warping method might not be a good way to preserve the structure of the object/hand. I think the authors could spend some space on this limitation and discuss possible ways for improvement.

---

> ### Author Response · Authors · 2020-11-20
> **Response to R3**
>
> Thank you for the constructive comments and suggestions. We address your comments below.
>
> **Q1: In the stage of pixel generation, it seems that the frame generation function cannot handle the overlap case very well. For example, in the "pick place" video in Figure 4, when the green cone is picked up and moved towards, it should have occluded the yellow cone at certain steps. However, it is occluded by the yellow one instead in the generated video. But since the masks  Mt−1,Mt, are given individually, I personally think it should not be the case: since there is only one object moving (i.e. green cone), you can warp that specific mask and simply stick that on top of the original frame. Can the authors explain why does it fail?.**
> In this work we present an hierarchical and modular pipeline for video synthesis. First, actions are scheduled for a specific timestep. Next, the scene layout is predicted. Finally, the future flow is predicted and refined. While this pipeline is fairly general, we believe these representations can be tweaked depending on the video data used. Specifically for the case of overlapping objects, we believe this is because the intermediate bounding box layout is too coarse. Instead, this intermediate representation can be replaced with a segmentation mask based representation, which will make the relation between the objects more explicit. We agree that this should be discussed and revise the Discussion accordingly.
>
> **Q2: The results on Something-Something indicate that flow-warping method might not be a good way to preserve the structure of the object/hand. I think the authors could spend some space on this limitation and discuss possible ways for improvement.?** In this work, we present a hierarchical and modular pipeline of video synthesis: first, the actions are scheduled for a specific timestep, then the scene layout is predicted, and finally, the future flow is predicted and refined. While this pipeline is reasonably general, more fine-grained intermediate representations could be used for improving the structure of objects. For example, to better address a hand synthesis case, a pose prediction step can be added before the flow prediction process. We agree that this should be discussed, and we revise the manuscript in the experiments section accordingly.

---

### Author Response · Authors · 2020-11-20
**Shared Comments**

**Shared Comments**
-------------------------------------------------------------------------------------------------------------------------------
In this work, we address goal-oriented video synthesis conditioned on a structure called Action Graphs. We consider the new task of generating a video that follows input instructions given in an Action Graph (AG), and we introduce a novel model for this task.

We are happy the reviewers found the proposed task **"well-motivated"** (R2, R4), **"thorough"** (R1) with **"clear explanations"** (R2), the three-level abstraction and formulation are **"neat and straightforward"**  (R3), while the generalization of novel compositions **"are interesting and show promising results"**  (R1, R2, R3). Moreover, the reviewers agreed that our method **"shows better performance"**  (R1, R2, R3, R4) and **"produces high-quality results"** (R2) in a difficult task.

Below we address the main points raised by reviewers, including how the approach would generalize to other datasets, applying Action Graphs to single action videos, and the task of collecting labels for new datasets. We also revised the manuscript accordingly.

---

### Decision · Program_Chairs · 2021-01-07
**Final Decision**

**Decision:**

Reject

**Comment:**

The paper focuses on the task of conditional video synthesis starting from a single image. The authors propose an *Action Graph* to model the configuration of objects, their interactions, and actions. They show promising results on two benchmark datasets (one synthetic and another realistic).

Based on the reviewers' comments and the limited discussion that ensued, it seems that some concerns in the paper were addressed by the authors; however, a main concern persists, namely the applicability of this Action Graph representation to more complex realistic videos (*e.g.* for datasets such as Kinetics and AVA). The authors do mention a manner in which an automated extraction of Action Graphs can be done, specifically with off-the-shelf (spatial or spatiotemporal) detectors for actions, objects, and object-object interactions. However, these are complicated tasks in their own right and still open problems in the field. Given that the Action Graph computable from this automated pipeline will undoubtedly contain noise (compounded by the errors of each component of this pipeline), the paper could have made a stronger argument for its contributions in realistic video, if for example an ablation study was done where *noisy* action graphs were used in training. Without more evidence that this representation will be applicable to more realistic scenarios of interest, it is difficult to gauge the impact it will have on the community. Despite its merits and promising initial results, the authors are encouraged to address this persisting concern and the other reviewers' comments to produce a stronger submission in the future.